# The Association of Free Fatty Acids and Eicosanoids with the Severity of Depressive Symptoms in Stroke Patients

**DOI:** 10.3390/ijms21155220

**Published:** 2020-07-23

**Authors:** Dariusz Kotlega, Agnieszka Zembron-Lacny, Monika Golab-Janowska, Przemyslaw Nowacki, Malgorzata Szczuko

**Affiliations:** 1Department of Neurology, Pomeranian Medical University Szczecin, 70-204 Szczecin, Poland; dkotlega@uz.zgora.pl (D.K.); monikagj@op.pl (M.G.-J.); przemyslaw.nowacki@pum.edu.pl (P.N.); 2Department of Applied and Clinical Physiology, Collegium Medicum University of Zielona Gora, 65-417 Zielona Gora, Poland; 3Department of Human Nutrition and Metabolomics, Pomeranian Medical University in Szczecin, 70-204 Szczecin, Poland; malgorzata.szczuko@pum.edu.pl

**Keywords:** anti-inflammatory action, docosahexaenoic acid, neuroinflammation, omega-3 polyunsaturated fatty acids

## Abstract

The study was designed to demonstrate the relationship of free fatty acids (FFAs) and eicosanoids levels with the severity of depressive symptoms in stroke. The ischemic stroke patients (*n* = 74) were included in the prospective study. The risk of depression was evaluated by the Beck Depression Inventory-II (BDI-II) 7 days and 6 months after the stroke onset. FFAs and inflammatory metabolites were determined by gas chromatography and liquid chromatography. In the acute phase of stroke, BDI-II and FFAs inversely correlated with C13:0 tridecanoic acid, C15:1 cis-10-pentadecanoid acid, C17:1 cis-10- heptadecanoid acid, C18:0 stearic acid, C20:3n6 eicosatrienoic acid, C22:1cis13 docosenoic acid and C22:6n3 docosahexaenoic acid (DHA). DHA level was significantly lower in patients with low vs. high BDI-II score. In the follow-up examination, BDI-II score directly correlated with C16:0 palmitic acid. The changes in BDI-II score during 6-month observation inversely correlated with lipoxin A4 and protectin D1, and directly correlated with 5-oxo-ETE. Importantly, the severity of depressive symptoms was associated with n3 PUFA level. Diet-derived FFAs were observed to potentially affect the inflammatory pathways in pathogenesis of depression in stroke and reduced DHA levels can attenuate depressive symptoms in stroke patients.

## 1. Introduction

Depression is a common disorder with lifetime prevalence ranging from 1.5% to 19%, with a probability of lifelong persistence reaching up to 30%. The most common scale for depressive symptoms assessment which is used by psychiatrists and psychologists is the Beck Depression Inventory (BDI) [1]. The Beck Depression Inventory-II is a screening tool for clinicians, and it is characterised by high test-reliability and acceptable internal consistency comparable to other tests. It is a useful tool in mental healthcare system applied to detect depressive symptoms [2]. There are two dominant theories of depression pathogenesis: Monoamine deficiency and neurogenesis disorder; however, inflammation has also been recently implicated in the origin of depression [3,4,5].

Ischemic stroke is the main cause of disability in adults. Additionally, it provokes neuropsychological deterioration and depression among patients. The prevalence of depression within 5 years after stroke varies between 39 to 52% [6,7]. Stroke patients in the acute phase develop depressive symptoms or depression in 5 to 54% of cases depending on the tools used for the depressive symptoms measurement. Depressive symptoms in the acute phase of stroke were found to be associated with the persistence of depression and increased mortality after one year [8].

The neurobiology of depression reveals that there is dysregulation of the central and peripheral immune system. In the course of depression, pathological changes can occur in brain microglia which, along with brain-resident macrophages, mediate neuroplasticity, and integrate neuroimmune signals. Neurons can activate microglia by the contact-dependent signals [9]. Chemokines attract immune cells to the sites with ongoing inflammation, and they are engaged in synaptic transmission, neuron–glia communication, neurogenesis and neuroinflammatory cross-talk [10,11]. Higher levels of chemokines and interleukins were observed in depressed subjects, which confirms the supposition that inflammation plays a central role in the pathogenesis of depression in stroke patients [11,12,13,14]. The immune system alterations observed in major depressive disorder (MDD) affect overall leukocyte level, absolute natural killer (NK)—cell counts, relative T-cell proportions, CD4/CD8 ratios, T-, and NK-cell function [15]. Following the inflammatory theory of depression pathogenesis, anti-inflammatory treatments have been prescribed. Acetylsalicylic acid (ASA), NSAIDs or statins application was connected with a decreased risk of early-onset depression after stroke, but ASA or NSAIDs were associated with an increase in the risk of depression after 12 months. The findings may indicate a different inflammatory pathomechanism in early and late-onset depression in stroke patients [16] The effect of ASA on COX-2 and inhibition of arachidonic acid (AA) cascade is worthy of note. Some authors agree with the presumptions that the increase in pro-inflammatory cytokines within days up to weeks following stroke may increase the risk of depression shortly after stroke, possibly through activation of indolamin 2.3-dioxygenase and synthesis of kynurenine instead of serotonin [17]. The anti-inflammatory treatment suppresses cytokine activity in the acute phase of stroke, and simultaneously reduces the severity of stroke [16]. Therefore, the use of drugs that modulate the immune response may have a different effect depending on the duration of post vascular incident period.

FFAs play a role in the synthesis of inflammatory molecules, neurotransmission and receptor function [18]. Polyunsaturated fatty acids are incorporated into cholesterol esters, phospholipid membranes and are used in the synthesis of prostaglandins and leukotriens. They have functional and structural role throughout the receptor function and cell membranes. The n3 α-linolenic acid and n6 linoleic acid are the essential fatty acids and must be delivered from the dietary sources. They are enzymatically modified and elongated into n6 arachidonic acid (AA), n3 eicosapentaenoic acid (EPA) and n3 docosahexaenoic acid [18]. The brain contains high levels of PUFAs (25–30%), mainly DHA (12–14% of total fatty acids), and AA (8–10% of total fatty acids). DHA represents 10–12% in astrocytes, 5% in the oligodendrocytes and up to 2% in microglial cells. The n3 PUFAs inhibitory activity affects several inflammatory pathways including leucocyte chemotaxis, adhesion molecule expression and leucocyte-endothelial adhesive interactions, production of eicosanoids like prostaglandins and leukotrienes from the n6 AA and production of pro-inflammatory cytokines. In addition, EPA gives rise to eicosanoids that often display a lower biological potency than those produced from AA [19]. Recent data show that n3 PUFAs exert an anti-inflammatory influence partly through the synthesis of specialized pro-resolving mediators (SPMs) such as resolvins, maresins, protectins, and their precursors such as 18R/S-HEPE, 17R/S-HDHA and 14R/S-HDHA [20,21]. The n3 DHA serves as precursor for D-series resolvins, protectins, and maresins, while the E-series resolvins come from n3 EPA, and lipoxins are derived from n6 AA [22]. Presumably, the role of n6 PUFA is mainly pro-inflammatory because AA is a precursor to pro-inflammatory mediators such as prostaglandins and leukotriens; however, an increased intake of AA does not elevate the levels of inflammatory markers [23]. Moreover, the AA rich oil may produce an anti-inflammatory effect by increasing the level of lipoxin A4 [24]. The systemic inflammation was reported to be related to diet quality [25]. The Western diet is associated with an increased level of CRP, Il-6, and E-selectin, while a healthy diet is characterized by a decreased level of CRP and E-selectin [26]. The Mediterranean diet is connected with a reduction in the level of CRP, Il-6, homocysteine, white blood counts, and fibrinogen [27]. Dietary omega-3 fatty acids are associated with plasma biomarker levels, reflecting lower levels of inflammation and endothelial activation in cardiovascular disease [28].

The levels of EPA, DHA, and total n3 PUFA were found to be significantly lower in depressive patients; however, no significant change has been recorded in AA or total n6 PUFA [29]. In a meta-review omega-3 was proven to have beneficial effects in the treatment of depression, with predominant effect of EPA formulas rather than DHA [30]. Another recent meta-analysis also showed beneficial impact of n3 fatty acids on depression symptoms but the effect was only observed in EPA formulations [31]. The anti-inflammatory effect of EPA can be responsible for the prevention of depression in hepatitis C virus patients treated with IFN-α [32]. The level of major trans fatty acid (TFA), elaidic acid (C18:1n-9t), was found to be associated with the risk of depression. Such a relation was confirmed by the intake of TFA measured with the dietary questionnaires [33,34]. The n6 PUFAs pro-inflammatory activity has been proven by the inducible COX-2 activation in the brain via the nuclear factor κB (NFκB) pathway. COX-2 catalyses the first step of the synthesis of thromboxanes and prostaglandins derived from n6 PUFAs that contribute to the initiation of inflammatory response [20]. At the same time, 15-HETE are produced through 15-LOX, transforming into proresolving mediators, such as lipoxins.

Some studies have also discussed the impact of saturated fatty acids (SFAs) on depression and reported their association with the inflammatory process modulation. SFAs may activate Toll-like receptor 4 (TLR4) signaling in adipocytes and macrophages and induce inflammatory response [35]. Palmitic acid and stearic acid induce a reactive microglial phenotype and increase the levels of inflammatory markers in a TLR4-dependent manner. The SFAs such as lauric, palmitic, and stearic acids induce NF-κB activation and expression of COX-2 and other inflammatory markers in macrophages [36,37].

The knowledge about the share of lifestyle factors, especially nutrition, in the pathomechanisms of stroke and depression is gradually verified. The ingestion of adequate quality food, and also the quantity of free fatty acids is essential to maintain an appropriate neuroinflammatory status. Therefore, we designed the study to show the relationships of FFA including the n3 PUFA and their inflammatory metabolites with the risk of depression in stroke patients.

## 2. Results

Analysing statistically significant differences between Beck 1 (BDI-II score on the 7^th^ day after stroke onset) and Beck 2 (BDI-II score after 6 months) patients for independent samples (all patients *n* = 74), no statistically significant differences were detected (Table 1). A similar result was observed when Beck 1 and Beck 2 groups were compared for dependent samples (patients participating in the study on day 7 and 6 months after the incident; *n* = 49) as shown in Table 1. The mean score of BDI-II in the acute phase of stroke was 9.311 (SD ± 7.034) in dependent groups and 9.980 (SD ± 7.207) in independent groups, while in the follow-up examination it amounted to 10.653 (SD ± 7.529) (Table 1).

Analysis of the correlation between Beck 1 in 74 patients and FFAs indicated several statistically significant relationships (Table 2). The increase in the number of points in the BDI-II scale correlated with the decrease in the levels of certain FFAs such as: C13:0 tridecanoic acid (r_s_ = −0.332), C15:1 cis-10-pentadecanoid acid (r_s_ = −0.297), C17:1 cis−10-heptadecanoid acid −0.251, C18:0 Stearic acid (r_s_ = −0.274), C20:3n6 eicosatrienoic acid (r_s_ = −0.234), C22:6n3 docosahexaenoic acid (r_s_ = −0.293) (Table 2).

The next step was to compare the relationships in a narrow group of patients with ischemic stroke to those who took part in the survey shortly after the incident and 6 months later (*n* = 49). When comparing the correlation between Beck 1, Beck 2, and delta Beck with FFA in a group of 49 patients, a significant negative relationship (Table 3) between Beck 1 with C13:0 tridecanoic acid (−0.315) and C22:6n3 docosahexaenoic acid (−0.295) was observed. Beck 2 (after 6 months of incident) was found to be correlated only with C16:0 palmitic acid (0.334). Interestingly, the delta Beck correlated directly with saturated acids like C16:0 palmitic acid (0.332), C18:0 stearic acid (0.395) and negatively with C18:1n9 ct oleic acid (−0.292) (Table 3).

After carrying out the FFAs analyses, selected pro and anti-inflammatory mediators, LA and AA derivatives were also determined (Table 4). Moreover, correlations were found in acute phase of stroke (Beck 1) with lipoxin A4 15-epi-LxA4 (0.446), while no correlation was found after 6 months (Beck 2). However, the correlation with the delta Beck was investigated and found to be negatively correlated with lipoxin A4 15-epi-LxA4 (−0.336) and protectin D1 (−0.368) while being directly correlated with 5-oxo-ETE (0.341) (Table 4).

When the whole group of 74 patients were analyzed in regard to the level of eicosanoids and BDI-II score in the acute phase of stroke, no significant correlation was found. Finally, patients were divided according to BDI-II scores into two groups: less (*n* = 55) and more advanced depressive symptoms (*n* = 19) with a cut-off at 14 points (Table 5 and Table 6).

In the comparative analysis of average percentages of FFA, significant differences in concentrations were observed only in regard to C22:6n3 docosahexaenoic acid (0.042) (Table 5). However, the analysis of FFA derivatives in both groups of patients did not show significant statistical differences (Table 6).

The associations between FFA, eicosanoids and subtypes of stroke according to the TOAST classification are presented in Table 7 and Table 8. The number of certain TOAST subgroups is as follows: 1 large-artery atherosclerosis (*n* = 27), 2 cardioembolism (*n* = 11), 3 small vessel occlusion, lacunar (*n* = 18), 4 other determined cause (*n* = 0) and 5 undetermined cause (*n* = 18).

## 3. Discussion

We detected associations between the level of free fatty acids and the score in BDI-II. This scale has been reported to demonstrate very good sensitivity and specificity [38]. Higher results indicate a greater severity of depressive symptoms, but there are large variances of cut-off scores for different populations, for instance 7 points for Parkinson’s disease patients or 10 points for depressive disorders in students [2,39]. Some authors suggest 0–12 score as an indicator of a minimal depression [40]. The BDI-II was performed in the acute phase of stroke and in the follow-up period of 6 months. The differences within the correlations of fatty acids and eicosanoids between these periods may be explained by the theory that stroke-induced increase in pro-inflammatory cytokines can lead to longer-term modification to the inflammatory process and can also modify the role of fatty acids in inflammation. Additionally, post-stroke patients usually have their chronic treatment introduced or changed and the pathomechanism of stroke is also connected with inflammation [41]. We did not analyze the effect of certain drugs received by the patients on the levels of FFAs and eicosanoids.

Studies on obesity and depression may be helpful in understanding the meaning of our findings. Obesity increases the risk of depression and adiponectin level is associated with the unfavorable course of bipolar disease [42,43]. It is proven that inflammatory markers are correlated with obesity [44]. SFA, such as palmitic acid, has been shown to induce activation of TLR4 receptors in hypothalamic microglia and to stimulate cytokine release indicating a potential mechanism by which high-fat diet leads to brain inflammation [45]. Notably, the hippocampus, a key brain region involved not only in learning and memory but also in depression and the response to antidepressants, is vulnerable to altered levels of cytokines, as they have important roles in synaptic plasticity and may inhibit neurogenesis [46,47]. This is particularly relevant to consider, given that the pathophysiology of depression may be distributed across several brain regions, including the hippocampus, hypothalamic-pituitary-adrenal axis, prefrontal cortex and striatum. This highlights the need to recognize dietary fat composition when studying factors that influence the etiology and the treatment of depression [48]. Metabolic disorders are also significant risk factors of stroke and other cardiovascular disorders. Obesity may provoke systemic basal low-grade inflammation and this can be a common pathogenetical factor of fatty acids, inflammation, stroke, and depression [49]. Higher amounts of fat intake can lead to lipopolysaccharide (LPS) diffusion from the gut to the circulatory system, which can activate inflammatory responses [50]. Subsequently, inflammation can mitigate development of depression depending on the severity and duration of inflammatory process. Such presumption can explain the direct associations between SFAs and MUFAs with the risk of depression, but we did not find an explanation for the inverse associations found in our study.

We detected a direct correlation between BDI-II and SFA C16:0 palmitic acid in the follow-up examination after 6 months. In the acute phase of stroke, the inverse association was observed in regard to SFA C13:0 tridecanoic acid, long-chain fatty acid C18:0 stearic acid and MUFA C17:1 cis-10- heptadecanoid acid. Other authors did not find any relation between these FFAs and depression [51]. Part of our results could be difficult to explain, especially that particular FFAs are related to diet in a different pattern: The level of C17:1 cis-10- heptadecanoid acid can increase in caloric reduction diet, while the level of C13:0 tridecanoic acid does not change [52]. Patients with MDD were shown to have increased blood level of palmitic acid in the recent meta-analysis [53]. Post-mortem analyses of brain tissue from MDD or bipolar disorder patients indicated an increase in palmitic acid [54,55]. These FFAs in the context of depression in stroke patients have not been analyzed yet. We did not detect any correlation between other FFAs (Table 1), which is consistent with other results regarding the risk of depression in patients without stroke, but the available data that would directly correspond to our analysis is lacking [56]. This is the first study to have analyzed such associations in stroke patients.

Clinical studies revealed that subjects diagnosed with depression or anxiety display significantly lower blood levels of n3 PUFAs and higher ratio of n6 to n3 PUFAs [18,57]. The n6 FFAs were also shown to have no effect on depression [58]. We observed the inverse association between DHA level and the risk of depression in both the matrix correlation in all patients and in the comparison of patients with low score (<14) vs. higher score (>13) in BDI-II. Such correlation was also detected in regard to C20:3n6 eicosatrienoic acid and BDI-II score in Beck 1 group of all patients. These findings are of note and are consistent with other results, although the efficacy in the treatment of depression was only proved in relation to EPA [29,30,31,59]. The therapeutical activity of EPA may result from the fact that EPA can inhibit phospholipase A2 and reduce the secretion of eicosanoids and pro-inflammatory cytokines, as depression pathogenesis is proposed to be associated with excessive secretion of pro-inflammatory cytokines [60,61]. The role of n3 PUFA in the pathogenesis of depression can result from the effect on the intracellular transmission, pre- and postsynaptic transmission or brain-derived neurotrophic factor (BDNF) synthesis [62]. Other authors suggest that the n3 and n6 PUFA impact on depression can be direct and unrelated to inflammation [63]. In a study on depression in acute coronary syndrome patients, similarly to our finding, a lower level of n3 PUFA and an unchanged level of n6 PUFA in depressed patients were detected [18]. In addition, the level of certain TFAs was reported to be positively associated with depressive symptoms [64]. No significant association between total n6 PUFA, C22:6n3 DHA and depression was found in other studies [29,56].

The inflammatory mediators of FFA mediate depression-like behavior induced by repeated social defeat stress through attenuating prefrontal dopaminergic activity, while resolvins attenuate neuroinflammation-associated depression [65]. In our study we observed an inverse correlation between n3-DHA and severity of depression, which could be explained by the anti-inflammatory effect of DHA. The supplementation of n3 PUFA increases also the level of precursors of SPMs such as 18-HEPE and 17-HDHA [21,66]. In this study we also observed an inverse correlation between change of BDI-II throughout 6 months (delta Beck) and the levels of SPMs such as lipoxin A4 and protectin D1 which both have anti-inflammatory properties. On the other hand, a direct association was observed in relation to pro-inflammatory AA metabolite-5-Oxo-ETE [67]. We did not find any correlation between other fatty acid inflammatory metabolites and the risk of depression in all the patients and in the comparison of patients with low score (<14) vs. higher score (>13) in BDI-II. To date, there have not been such studies performed in stroke patients. It has been suggested that inflammatory metabolites of FFAs take part in the regulation of physiological responses and pathogenesis of psychiatric disorders. The level of lipoxin A4 was negatively correlated with the score of hospital anxiety and depression in coronary heart disease patients [68]. The add-on therapy of COX-2 inhibitor has been demonstrated to decrease the severity of depression [69] and antidepressant activity of resolvins E1 and E2 have also been reported [70]. On the other hand, there has been no association found between depression and control groups in regard to leucotriene B4 [51]. Lipoxins and resolvins act in a stereospecific manner on distinct cell types through interaction with G-protein coupled receptors (GPCRs) to stimulate non-phlogistic macrophage phagocytosis, increase anti-inflammatory cytokines and decrease pro-inflammatory cytokine generation in macrophages, neutrophils, endothelial cells, and dendritic cells. Lipoxins and resolvins also stimulate endothelial production of nitric oxide (NO) and vasoprotective prostacyclin (PGI_2_) [71]. Protectin D1 effectively inhibits T cell migration and apoptosis and reduces the production and release of TNFα and Interferon gamma, besides limiting neutrophil activity and promoting macrophage activation [72].

An alternative explanation of the anti-inflammatory effect of n3 PUFA includes the altered cell membrane phospholipid fatty acid composition, a disruption of lipid rafts and inhibition of the pro-inflammatory transcription NFκB and subsequent activation of anti-inflammatory transcription factor peroxisome proliferator-activated receptor γ (PPARγ) [19]. We did not observe any association with 12-HETE and 15-HETE that promote the activation of PPARγ, which has neuroprotective functions through its anti-inflammatory properties [20].

We measured the levels of fatty acids in the serum but they reflect the levels in tissues, including the brain [29,73]. The association between the level of fatty acids and the risk of depression elucidated in our study did not provoke any significant change of their inflammatory metabolites. This result is of note and must be taken into account in further studies on this topic. Fatty acids can also play an inflammation-unrelated role in development of depression and stroke patients can constitute a pathogenetically diverse group compared to non-stroke patients, especially that this is the first study that analyzed such associations in stroke patients. There can also be confounding, metabolic factors affecting the role of fatty acids in inflammation and depression in stroke patients, such as the link of obesity with hypothalamic pituitary adrenal axis, abnormalities in BDNF signaling, adipose-derived hormones, insulin signaling or oxidative stress pathways [74,75].

We evaluated the association between the types of stroke according to the TOAST classification system, eicosanoids and FFAs. The level of maresin 1, which belongs to the SPM, was higher in embolic strokes compared to the large-artery subtype. Maresin 1 attenuates inflammation in endothelial cells and prevents progression of atherosclerosis, however, no relevant research is available to compare [76]. The proinflammatory lipid mediator leukotriene B4 reached a higher level in small-vessel type of stroke compared to a large-vessel one. The genetic variant of the leukotriene B4 receptor complex was shown to affect the risk of cardioembolic stroke [77]. The bioactive form of lipoxin A4-15-epi-lipoxin A_4_ (15-epi LXA_4_) that promotes resolution of inflammation by inhibiting superoxide generation and polymorphonuclear leukocyte transmigration was found to be higher in embolic when compared to undetermined subtype of stroke [78]. It was observed that higher levels of SFA and n6 PUFA were associated with lacunar and atherosclerotic stroke, while no relationships regarding the n3 DHA and the subtypes of stroke were observed [79,80]. Higher levels of C14:0 myristic acid were identified in small-vessel subtypes of stroke in comparison to large-vessel and embolic ones, but in other SFAs (C15:0 pentadecanoid acid) its level was lower in small-vessel compared to undetermined type. We observed consistent results in regard to n6 FFA—lower levels were detected in embolic strokes compared to large and small-vessel subtypes. These finding are all the more interesting that C20:3n6 eicosatrienoic acid, also known as dihomo-γ-linolenic acid (DGLA) is engaged in anti-inflammatory, anti-platelet, anti-thrombotic, and anti-atherogenic activities [81,82]. There is no direct data regarding the subtypes of stroke and the topic of our study. Bearing in mind the immunomodulating effects of certain FFAs and eicosanoids, we cannot exclude that the results obtained in the TOAST analysis may be of importance in the theory of common inflammatory background and interactions between lipids, their metabolites, stroke, and depression. Such aspects need further studies which should aim at assessing the role of FFAs and eicosanoids in the stroke pathomechanism.

## 4. Material and Methods

### 4.1. Subjects

Seventy-four ischemic stroke patients were included in the prospective study. The inclusion criterion was the ischemic stroke diagnosed on the basis of clinical symptoms and additional tests results, with standard treatment and brain imaging (computed tomography or nuclear magnetic resonance) [83,84]. Patients with both atherothrombotic and embolic stroke were also enrolled in the study. Stroke was defined as a syndrome of rapidly developing symptoms of focal or global cerebral dysfunction lasting ≥24 h or leading to death, with apparent vascular cause [85]. The stroke etiology was classified according to the TOAST classification system. The TOAST classification describes stroke subtypes as follows: (1) Large-artery atherosclerosis, (2) cardioembolism, (3) small vessel occlusion (lacunar), (4) other determined cause, and (5) undetermined cause [86]. The exclusion criteria included intracranial hemorrhage visible in brain imaging, symptoms of active infection including body temperature of more than 37.4 °C, clinical or biochemical symptoms of infection, active autoimmune disorder or malignancy as well as the speech or consciousness disturbances to enable reliable results of BDI-II testing. All subjects had FFA gas chromatography and inflammatory metabolites measurements performed with the use of liquid chromatography. The patients were hospitalized in the Neurology Department in the district hospital in Poland. All subjects were Caucasian. All patients were evaluated for the risk of depression with the Beck Depression Inventory-II (BDI-II) 7 days and 6 months after the stroke onset. The first assessment was performed in a group of 74 patients (Beck 1), while at the follow-up visit 49 patients were tested (Beck 2). The smaller number of patients on Beck 2 examination resulted from their failure to report for a follow-up visit. The cut-off point of the BDI-II scale was set at <14 points as patients with low score (*n* = 55) vs. those with >13 points as patients with greater BDI-II score (*n* = 19). All patients were Caucasians, of the mean age of 60.7 years (min. 14, max. 83), 33 males (44.6%), 61 patients with hypertension (82.4%), 36 patients with diabetes or impaired fasting glucose (48.6%), 8 patients suffered from coronary heart disease (10.8%), BMI ≥25 was recorded in 59 patients (79.7%), 26 were smokers (33.1%). Seven patients received intravenous thrombolytic infusion of alteplase, 11 patients were taking L-thyroxine, 61 patients were taking hipotensives, 17 were on hypoglicemics. All patients received acetylsalicylic acid and statins. All the subjects were informed of the aim of the study and signed a written consent to participate in the project. The protocol of the study was approved by the local Ethics Committee in Zielona Góra (N° 08/73/2017, 27 February 2017), in accordance with the Helsinki Declaration.

### 4.2. Risk of Depression Assessment

Participants filled in the 21-item BDI-II questionnaire by rating their answers on a scale from 0 to 3 points according to how they had been feeling over the last two weeks. The scale is consistent with DSM-IV criteria for major depression [87]. Total score ranges from 0 to 63 points. Higher results indicate a greater severity of depressive symptoms [2]. The patients completed the questionnaire during their hospitalization, 7 days after admission, and at the follow-up visit 6 months later.

### 4.3. Blood Collection

The fasting, venous blood was obtained with the use of Vacutainer system on the seventh day after the onset. Samples for the analysis of FFA and inflammatory mediators were centrifuged and stored in −80 Celsius degrees temperature until laboratory analysis was performed.

### 4.4. Fatty Acids Detection

Fatty acids methyl esters were isolated from serum with the use of the modified Folch and Szczuko methods [88,89]. The fatty acids profile was labelled by gas chromatography. The gas chromatography (GC) was performed with the use of the Agilent Technologies 7890A GC System (SUPELCOWAX™)10 Capillary GC Column (15 mm × 0.10 mm, 0.10 μm); Supelco (Bellefonte, PA, USA). Chloroform and methanol (Merck), boron trifluoride in methanol, NaCl and 2,6-Di-tert-butyl-4-methylphenol (Sigma-Aldrich, St Louis, MO, USA), double-distilled water (Milli-Q Water System, Millipore, Billerica, MA, USA), fatty acids standards were supplied by Sigma-Aldrich (Darmstadt, Germany), Cayman Chemical (Ann Arbor, MI, USA) and Neochema (Bodenheim, Germany). The results of the fatty acids were presented as the percentage distribution.

### 4.5. Chemicals and Reagents for Detection of Fatty Acids Metabolites

Methanol and acetic acid as mobile phases were used of liquid chromatography (HPLC), double-distilled water and buffers used for HPLC, as described above. Analyses were filtered through 0.22 µm nylon filters.

### 4.6. Sample Preparation

The following inflammatory mediators were analyzed: resolvin D1, maresine1, prostaglandin B2, 5(S),6(R), 15(R)-lipoxin A4, 5(S),6(R)-lipoxin A4, leucotriene B4, 16(RS)HEPE, 5(S)-HETE, 12(S)-HETE, 15(S)-HETE, 5(S)-oxoETE, 10(S)17(R)DiDHA, 16(R)/16(S)-HETE, 9(S)-HODE, 13(S)-HODE, 17(RS)HDHA. All derivatives were extracted from the 0.5ml of plasma by using a solid-phase extraction RP-18 SPE columns (Agilent Technologies, Santa Clara, CA, USA).

### 4.7. Instrumentation and Software

The HPLC separations were performed on the 1260 liquid chromatograph (Agilent Technologies, Santa Clara, CA, USA), degasser model G1379B, bin pump model G1312B, column oven model G1316A and G1315CDAD VL+. Samples were injected using model G1329B. The Agilent ChemStation software (Agilent Technologies, Cheadle, UK) was used for data acquisition, instrument control and analysis. The temperature of column oven was set at 210C. The separation was completed on Thermo Scientific Hypersil BDS C18 column 100 × 4.6 mm 2.4 µm.

### 4.8. HPLC Operating Parameters

A gradient method was used, with mixture of solvent A (methanol/water/acetic acid, 50/50/0.1, *v*/*v*/*v*) and B (methanol/water/acetic acid, 100/0/0.1, *v*/*v*/*v*) that constituted the mobile phase. The buffer B percentage in the mobile phase was 30% at 0.0 min to 2.00 min of separation, increased linearly to 80% at 33 min, 98% (33.1–37.5 min) and 30% (40.3–45 min). The flow rate was 1.0 mL/min. The sample injection volume was 60 μL. The DAD detector monitored peaks by adsorption at 235 nm for 16(RS)-HEPE, 17(RS)-HDHA, 9(S)-HODE, 13(S)-HODE, 5(S)-HETE, 12(S) HETE and 15S HETE, at 280 nm for PGB2 (prostaglandin B2, internal standard) and 5(S)- oxoETE, resolvin E1, 10(S)17(R)- DiDHA, maresine1, leucotriene B4 at 210 nm for prostaglandin E2, 16(R)-HETE and 16(S)-HETE (the two latter were eluted as one peak) and at 302 nm for 5(S),6(R)-lipoxin A4, 5(S),6(R), 15(R)-lipoxin A4 and resolvin D1. To confirm the identification of analytes, absorbance spectra of peaks were analyzed. The quantitation was based on peak areas with internal standard calibration. ChemStation Software was used for the quantitative analysis (Agilent Technologies, Cheadle, UK).

### 4.9. Statistical Analysis

Statistical analyses were performed by means of statistical software Statistica 13.1 (StatSoft Inc., Tulsa, OK, USA). The assumptions for the use of parametric or non-parametric tests were checked using the Shapiro–Wilk to evaluate the normality of the distributions. The correlation matrix was made/performed for the combined BDI-II score with FFA and for BDI-II score with inflammatory mediators using Spearman’s rank correlation (r_s_—Spearman rank correlation coefficient) in Beck1 and Beck 2 groups. Statistical significance was set at *p* < 0.05. The results are expressed as mean and standard deviation (x ± SD).

## 5. Conclusions

While emphasizing the importance of nutrition itself, we also contributed to the nutritional and subsequently inflammatory theory of depression development in general population and in stroke patients in particular. For the first time we elucidated a potential common association between dietary and inflammatory factors as those that affect the risk of depression with regard to the inflammatory theory of stroke pathomechanism. The severity of depressive symptoms, measured by means of BDI-II scale, is clearly related to the arachidonic acid cascade directly after the stroke onset and 6 months later. Moreover, the low level of DHA, in particular PUFA n3, can be a potential factor affecting the risk of depression and severity of depressive symptoms in stroke patients.

## 6. Limitations

The limitation of our study includes the fact that we performed the follow-up examination of subjects after 6 months in regard to the BDI-II scale but we did not provide follow-up assessment with the NIHSS scale. The severity and risk of depression may be related to the degree of neurological deficit. The improvement in the functional outcome could affect the BDI-II score in the follow-up period. It would be interesting to assess such potential relationships.

## Figures and Tables

**Table 1 ijms-21-05220-t001:** Comparison of Beck Depression Inventory-II (BDI-II) score obtained on 7th day (Beck 1) and in follow-up after 6 months (Beck 2).

	Beck 1	Beck 2	*p*-Value
Independent tests *n* = 74	9.98 ± 7.207	10.653 ± 7.529	0.315
Dependent tests *n* = 49	9.311 ± 7.034	10.653 ± 7.529	0.415

**Table 2 ijms-21-05220-t002:** Association between free fatty acids (FFAs) and BDI-II score obtained on 7th day (Beck 1) and comparison of free fatty acids (FFAs) mean values between groups.

FFA [%]	Correlation Matrix at Stroke Onset Beck 1 *n* = 74 r_s_-Value	Mean [%] ± SD; Beck 1 *n* = 49	Mean [%] ± SD; Beck 1 *n* = 74	*p*-Value
C13:0 tridecanoic acid	−0.332 *	0.299 ± 0.096	0.307 ± 0.092	0.62
C14:0 myristic acid	0.008	1.192 ± 0.328	1.208 ± 0.381	0.81
C14:1 myristolenic acid	0.064	0.070 ± 0.037	0.070 ± 0.038	0.99
C15:0 pentadecanoid acid	−0.120	0.182 ± 0.063	0.217 ± 0.108	0.04
C15:1 cis-10-pentadecanoid acid	−0.297 *	0.072 ± 0.030	0.081 ± 0.036	0.16
C16:0 palmitic acid	0.003	26.506 ± 1.670	26.802 ± 1.742	0.35
C16:1 palmitoleic acid	0.166	2.216 ± 0.843	2.134 ± 0.747	0.57
C17:0 heptadecanoic acid	−0.173	0.291 ± 0.045	0.302 ± 0.050	0.23
C17:1 cis-10- heptadecanoid acid	−0.251 *	0.084 ± 0.035	0.091 ± 0.035	0.27
C18:0 stearic acid	−0.274 *	13.046 ± 2.037	13.314 ± 1.983	0.47
C18:1n9 ct oleic acid	0.086	22.606 ± 4.158	22.591 ± 3.713	0.98
C18:1 vaccinic acid	0.031	1.946 ± 0.346	1.978 ± 0.351	0.61
C18:2n6c linoleic acid	−0.016	11.392 ± 2.365	11.538 ± 2.333	0.74
C18:2n6t linoleic acid	0.158	6.702 ± 1.581	6.141 ± 1.931	0.09
C18:3n6 gamma linoleic acid	0.100	0.421 ± 0.204	0.386 ± 0.192	0.34
C18:3n3 linolenic acid	−0.0139	0.511 ± 0.156	0.504 ± 0.159	0.81
C18:4 stearidonate	−0.031	0.060 ± 0.028	0.057 ± 0.027	0.56
C20:0 arachidic acid	0.039	0.217 ± 0.084	0.206 ± 0.073	0.46
C22:1/C20:1 Cis11- eicosanic acid	−0.015	0.177 ± 0.079	0.179 ± 0.069	0.93
C20:2 Cis-11-eicodienoic acid	0.046	0.149 ± 0.037	0.151 ± 0.034	0.81
C20:3n6 eicosatrienoic acid	−0.234 *	1.316 ± 0.329	1.283 ± 0.309	0.57
C20:4n6 arachidonic acid	−0.046	6.335 ± 1.363	6.305 ± 1.314	0.90
C20:3n3 Cis-11-eicosatrienoic acid	0.115	0.030 ± 0.014	0.031 ± 0.014	0.88
C20:5n3 eicosapentaenoic acid	−0.095	0.620 ± 0.260	0.603 ± 0.258	0.73
C22:0 behenic acid	0.179	0.248 ± 0.102	0.225 ± 0.098	0.21
C22:1n9 13 erucic acid	−0.035	0.065 ± 0.016	0.037 ± 0.016	0.33
C22:2 cis-docodienoic acid	0.002	0.017 ± 0.031	0.022 ± 0.027	0.58
C23:0 tricosanoic acid	−0.019	0.219 ± 0.012	0.017 ± 0.011	0.79
C22:4n6 docosatetraenoate	−0.107	0.195 ± 0.095	0.233 ± 0.152	0.57
C22:5w3 ddocosapentaenate	−0.002	0.483 ± 0.080	0.223 ± 0.116	0.15
C24:0 lignoceric acid	0.178	0.171 ± 0.275	0.460 ± 0.231	0.63
C22:6n3 docosahexaenoic acid	−0.293 *	1.702 ± 0.079	0.153 ± 0.076	0.19
C24:1 nervonic acid	0.201	0.460 ± 0.564	1.752 ± 0.531	0.62

* *p* < 0.05 statistically significant correlation matrix (Spearman’s rank correlation).

**Table 3 ijms-21-05220-t003:** The association between FFA and BDI-II score obtained on 7th day (Beck 1) and in follow-up after 6 months (Beck 2), and delta values.

FFA [%]	Beck 1 *n* = 49 r_s_-Value	Beck 2 *n* = 49 r_s_-Value	Delta Beck
C13:0 tridecanoic acid	−0.315 *	−0.098	0.276
C14:0 myristic acid	0.185	0.188	0.015
C14:1 myristolenic acid	0.180	0.205	0.046
C15:0 pentadecanoid acid	−0.013	0.103	0.157
C15:1 cis-10-pentadecanoid acid	−0.274	−0.066	0.263
C16:0 palmitic acid	0.081	0.334 *	0.332 *
C16:1 palmitoleic acid	0.148	0.083	−0.072
C17:0 heptadecanoic acid	−0.038	0.129	0.228
C17:1 cis-10- heptadecanoid acid	−0.213	−0.030	0.234
C18:0 stearic acid	−0.205	0.106	0.395 *
C18:1n9 ct oleic acid	0.138	−0.095	−0.292 *
C18:1 vaccinic acid	0.037	−0.138	−0.221
C18:2n6c linoleic acid	−0.086	−0.099	−0.036
C18:2n6t linoleic acid	0.042	0.101	0.088
C18:3n6 gamma linoleic acid	0.114	−0.029	−0.179
C18:3n3 linolenic acid	0.019	−0.141	−0.203
C18:4 stearidonate	0.050	−0.091	−0.177
C20:0 arachidic acid	−0.018	−0.101	−0.108
C22:1/C20:1 Cis11- eicosanic acid	−0.010	−0.186	−0.229
C20:2 Cis−11-eicodienoic acid	0.048	0.116	0.093
C20:3n6 eicosatrienoic acid	0.063	0.116	0.067
C20:4n6 arachidonic acid	−0.133	−0.138	−0.013
C20:3n3 Cis-11-eicosatrienoic acid	−0.207	−0.124	0.100
C20:5n3 eicosapentaenoic acid	−0.076	−0.136	−0.075
C22:0 behenic acid	0.062	0.036	−0.033
C22:1n9 13 erucic acid	−0.066	0.009	0.095
C22:2 cis-docodienoic acid	−0.036	0.145	0.239
C23:0 tricosanoic acid	0.083	0.015	−0.091
C22:4n6 docosatetraenoate	−0.117	0.018	0.174
C22:5w3 docosapentaenate	−0.013	−0.172	−0.203
C24:0 lignoceric acid	0.054	0.045	−0.014
C22:6n3 docosahexaenoic acid	−0.295 *	−0.193	0.124
C24:1 nervonic acid	0.075	−0.054	−0.170

* *p* < 0.05 statistically significant correlation matrix (Spearman’s rank correlation).

**Table 4 ijms-21-05220-t004:** The association between inflammation mediators and BDI-II score obtained on 7th day (Beck 1) and in follow-up after 6 months (Beck 2), and delta values.

Eicosanoids [µg/mL]	Beck 1 *n* = 49 r_s_-Value	Beck 2 *n* = 49 r_s_-Value	Delta Beck
Resolvin E1	0.021	0.076	0.075
Prostaglandin E2	0.090	−0.032	−0.151
Resolvin D1	0.007	−0.101	−0.138
Lipoxin A4 LxA4 5S, 6R	−0.141	−0.071	0.086
Lipoxin A4 15-epi-LxA4 A4 5S, 6R, 15R	0.446	0.176	−0.336
Protectin D1	0.232	−0.059	−0.368
Maresin 1	−0.036	0.141	0.231
Leucotriene B4	0.012	0.054	0.060
18RS HEPE	−0.110	−0.005	0.134
16RS HETE	−0.151	−0.205	−0.078
13S HODE	0.041	0.014	−0.028
9S HODE	−0.045	0.039	0.112
15S HETE	0.045	0.110	0.086
17RS HDHA	0.109	0.116	0.017
12S HETE	−0.057	−0.104	−0.060
5-oxo-ETE	−0.064	0.193	0.341
5 HETE	0.058	0.041	−0.014

* *p* < 0.05 statistically significant correlation matrix (Spearman’s rank correlation).

**Table 5 ijms-21-05220-t005:** The comparison of free fatty acids level between patients with low (<14) vs. higher (>13) BDI-II score.

FFA [%]	Mean [%] ± SD in Patients with Low BDI-II Score; *n* = 55	Mean [%] ± SD in Patients with Higher BDI-II Score; *n* = 19	*p*-Value
C13:0 tridecanoic acid	0.317 ± 0.087	0.284 ± 0.098	0.17
C14:0 myristic acid	1.197 ± 0.364	1.259 ± 0.435	0.542
C14:1 myristolenic acid	0.068 ± 0.034	0.076 ± 0.048	0.439
C15:0 pentadecanoid acid	0.223 ± 0.116	0.204 ± 0.079	0.497
C15:1 cis-10-pentadecanoid acid	0.085 ± 0.039	0.071 ± 0.023	0.131
C16:0 palmitic acid	26.873 ± 1.773	26.621 ± 1.725	0.592
C16:1 palmitoleic acid	2.106 ± 0.705	2.265 ± 0.856	0.426
C17:0 heptadecanoic acid	0.307 ± 0.053	0.289 ± 0.04	0.206
C17:1 cis-10- heptadecanoid acid	0.094 ± 0.037	0.083 ± 0.027	0.226
C18:0 stearic acid	13.591 ± 1.89	12.735 ± 1.963	0.096
C18:1n9 ct oleic acid	22.310 ± 3.519	23.219 ± 4.267	0.362
C18:1 vaccinic acid	1.954 ± 0.34	2.047 ± 0.387	0.324
C18:2n6c linoleic acid	11.498 ± 2.457	11.690 ± 2.05	0.76
C18:2n6t linoleic acid	5.997 ± 1.905	6.393 ± 1.941	0.439
C18:3n6 gamma linoleic acid	0.39 ± 0.1996	0.385 ± 0.174	0.93
C18:3n3 linolenic acid	0.498 ± 0.139	0.493 ± 0.179	0.908
C18:4 stearidonate	0.058 ± 0.028	0.056 ± 0.026	0.786
C20:0 arachidic acid	0.204 ± 0.075	0.212 ± 0.071	0.722
C22:1/C20:1 Cis11- eicosanic acid	0.175 ± 0.067	0.184 ± 0.072	0.613
C20:2 Cis-11-eicodienoic acid	0.149 ± 0.03	0.152 ± 0.044	0.743
C20:3n6 eicosatrienoic acid	1.276 ± 0.315	1.317 ± 0.298	0.616
C20:4n6 arachidonic acid	6.376 ± 1.252	6.082 ± 1.524	0.408
C20:3n3 Cis-11-eicosatrienoic acid	0.032 ± 0.016	0.027 ± 0.009	0.176
C20:5n3 eicosapentaenoic acid	0.628 ± 0.271	0.541 ± 0.215	0.211
C22:0 behenic acid	0.22 ± 0.095	0.242 ±0.108	0.391
C22:1n9 13 erucic acid	0.037 ± 0.017	0.037 ± 0.011	0.98
C22:2 cis-docodienoic acid	0.017 ± 0.01	0.018 ± 0.011	0.691
C23:0 tricosanoic acid	0.241 ± 0.17	0.215 ± 0.082	0.525
C22:4n6 docosatetraenoate	0.231 ± 0.123	0.201 ± 0.099	0.348
C22:5w3 docosapentaenate	0.471 ± 0.263	0.429 ± 0.109	0.505
C24:0 lignoceric acid	0.149 ± 0.077	0.164 ± 0.076	0.458
C22:6n3 docosahexaenoic acid	1.823 ± 0.529	1.534 ± 0.499	0.042
C24:1 nervonic acid	0.38 ± 0.224	0.455 ± 0.308	0.255

* *p* < 0.05 statistically significant differences.

**Table 6 ijms-21-05220-t006:** The comparison of eicosanoids level between patients with low (<14) vs. higher (>13) BDI-II score.

Eicosanoids [µg/mL]	Mean ± SD in Patients with Lower BDI-II Score *n* = 55	Mean ± SD in Patients with higher BDI-II Score *n* = 19	*p*-Value
Resolvin E1	0.054 ± 0.088	0.070 ± 0.099	0.504
Prostaglandin E2	3.319 ± 3.14	3.185 ± 6.508	0.906
Resolvin D1	0.174 ± 0.267	0.171 ± 0.22	0.969
Lipoxin A4 LxA4 5S, 6R	0.023 ± 0.171	0.00	0.56
Lipoxin A4 15-epi-LxA4 A4 5S, 6R, 15R	0.018 ± 0.037	0.036 ± 0.053	0.109
Protectin D1	0.045 ± 0.053	0.053 ± 0.09	0.629
Maresin 1	0.031 ± 0.016	0.030 ± 0.016	0.717
Leukotriene B4	0.025 ± 0.014	0.029 ± 0.014	0.323
18RS HEPE	0.111 ± 0.038	0.104 ± 0.035	0.490
16RS HETE	0.014 ± 0.078	0.00	0.447
13S HODE	0.032 ± 0.031	0.033 ± 0.025	0.856
9S HODE	0.034 ± 0.03	0.03 ± 0.02	0.641
15S HETE	0.292 ± 0.217	0.294 ± 0.17	0.971
17RS HDHA	0.124 ± 0.092	0.118 ± 0.06	0.796
12S HETE	1.787 ± 1.143	1.78 ± 1.121	0.98
5-oxo-ETE	0.186 ± 0.103	0.187 ± 0.096	0.971
5 HETE	0.025 ± 0.014	0.025 ± 0.011	0.996

* *p* < 0.05 statistically significant differences.

**Table 7 ijms-21-05220-t007:** The comparison of subgroups (*n* = 74) in regard to TOAST classification and FFA.

FFA [%]	1 vs. 2	1 vs. 3	1 vs. 5	2 vs. 3	2 vs. 5	3 vs. 5
C13:0 tridecanoic acid	NS	NS	NS	NS	NS	NS
C14:0 myristic acid	NS	0.033	NS	0.041	NS	NS
C14:1 myristolenic acid	NS	NS	NS	NS	NS	NS
C15:0 pentadecanoid acid	NS	NS	NS	NS	NS	0.021
C15:1 cis-10-pentadecanoid acid	NS	NS	NS	NS	NS	NS
C16:0 palmitic acid	NS	NS	NS	NS	NS	NS
C16:1 palmitoleic acid	NS	NS	NS	NS	NS	NS
C17:0 heptadecanoic acid	NS	NS	NS	NS	NS	NS
C18:0 stearic acid NS	NS	NS	NS	NS	NS	NS
C18:1n9 ct oleic acid	NS	NS	NS	NS	NS	NS
C18:1 vaccinic acid	NS	NS	NS	NS	NS	NS
C18:2n6c linoleic acid	NS	NS	NS	NS	NS	NS
C18:2n6t linoleic acid	NS	NS	NS	NS	NS	NS
C18:3n6 gamma linoleic acid	0.017	NS	0.037	0.025	NS	NS
C18:3n3 linolenic acid	NS	NS	0.006	NS	NS	NS
C18:4 stearidonate	NS	NS	NS	NS	NS	NS
C20:0 arachidic acid	NS	NS	NS	NS	NS	0.036
C22:1/C20:1 Cis11- eicosanic acid	NS	NS	NS	NS	NS	NS
C20:2 cis-11-eicodienoic acid	NS	NS	NS	NS	NS	NS
C20:3n6 eicosatrienoic acid	<0.001	NS	NS	0.007	0.007	NS
C20:4n6 arachidonic acid	NS	NS	NS	NS	NS	NS
C20:3n3 Cis-11-eicosatrienoic acid	NS	NS	NS	NS	NS	NS
C20:5n3 eicosapentaenoic acid	NS	NS	NS	NS	NS	0.034
C22:0 behenic acid	NS	NS	NS	NS	NS	0.034
C22:1n9 13 erucic acid	NS	NS	NS	NS	NS	NS
C22:2 cis-docodienoic acid	NS	NS	NS	NS	NS	0.039
C23:0 tricosanoic acid	NS	NS	NS	NS	NS	NS
C22:4n6 docosatetraenoat	NS	NS	NS	NS	NS	NS
C22:5w3 ddocosapentaenate	NS	NS	NS	NS	NS	NS
C24:0 lignoceric acid	NS	NS	NS	NS	NS	0.034
C22:6n3 docosahexaenoic acid	NS	NS	NS	NS	NS	NS
C24:1 nervonic acid	NS	NS	NS	NS	NS	NS

NS statistically non-significant differences, *p* < 0.05 statistically significant differences, TOAST classification: 1 large-artery atherosclerosis, 2 cardioembolism, 3 small vessel occlusion (lacunar), 5 undetermined cause.

**Table 8 ijms-21-05220-t008:** The comparison of subgroups (*n* =74) in regard to TOAST classification and eicosanoids.

Eicosanoids [µg/mL]	1 vs. 2	1 vs. 3	1 vs. 5	2 vs. 3	2 vs. 5	3 vs. 5
Resolvin E1	NS	NS	NS	NS	NS	NS
Prostaglandin E2	NS	NS	NS	NS	NS	NS
Resolvin D1	NS	NS	NS	NS	NS	NS
Lipoxin A4 LxA4 5S, 6R	NS	NS	NS	NS	NS	NS
Lipoxin A4 15-epi-LxA4 A4 5S, 6R, 15R	NS	NS	NS	NS	0.028	0.011
Protectin D1	NS	NS	NS	NS	NS	NS
Maresin 1	0.019	NS	NS	NS	NS	NS
Leucotriene B4	NS	0.037	NS	NS	NS	NS
18RS HEPE	NS	NS	NS	NS	NS	NS
16RS HETE	NS	NS	NS	NS	NS	NS
13S HODE	NS	NS	NS	NS	NS	NS
9S HODE	NS	NS	NS	NS	NS	NS
15S HETE	0.011	NS	NS	NS	NS	NS
17RS HDHA	NS	NS	NS	NS	NS	NS
12S HETE	NS	NS	NS	NS	NS	NS
5-oxo-ETE	NS	NS	0.045	NS	NS	NS
5 HETE NS	NS	NS	NS	NS	NS	NS

NS statistically non-significant differences, *p* < 0.05 statistically significant differences, TOAST classification: 1 large-artery atherosclerosis, 2 cardioembolism, 3 small vessel occlusion (lacunar), 5 undetermined cause.

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
