# Peer review of "The Association of Free Fatty Acids and Eicosanoids with the Severity of Depressive Symptoms in Stroke Patients"

_ijms, 2020, doi:10.3390/ijms21155220_

Round 1

Reviewer 1 Report

This study aimed to study the relationship of free fatty acids (FFAs) and eicosanoids levels with the severity of depressive symptoms in stroke. Generally, the study raised an interesting and important point in stroke area. But there are some points that the authors should clarify before further consideration.

  1. The criteria for the inclusion of stroke patients should be placed. The authors only describe “The inclusion criterion was the ischemic stroke diagnosed on the basis of clinical symptoms and additional tests results, with standard treatment” in 4.1 Subjects section. More details should be added to avoid over-interpretation.
  2. The depression symptom may be affected by stroke type. For example, Toast classification, stroke of atherosclerosis or cardioembolism? Would that influence the results? The authors should give subgroup analysis?
  3. NIHSS is a commonly used scale to assess stroke severity. Patients with severe symptom may be more likely to fall into a depressive state. And did they get any improvement during follow-up? The authors should clarify this.
  4. More “standard treatment” information should be provided.

Author Response

Review 1

We greatly appreciate your time and effort dedicated to providing feedback on our manuscript and we are grateful for the insightful comments on and valuable improvements to our paper. All the suggestions helped us to evaluate our outcomes even more precisely in order to deliver improved, high quality scientific manuscript which we hope will now meet the high standards of International Journal of Molecular Sciences.

Open Review

English language and style

( ) Extensive editing of English language and style required 
( ) Moderate English changes required 
(x) English language and style are fine/minor spell check required 
( ) I don't feel qualified to judge about the English language and style

Yes

Can be improved

Must be improved

Not applicable

Does the introduction provide sufficient background and include all relevant references?

( )

( )

(x)

( )

Is the research design appropriate?

( )

(x)

( )

( )

Are the methods adequately described?

( )

( )

(x)

( )

Are the results clearly presented?

( )

( )

(x)

( )

Are the conclusions supported by the results?

( )

( )

(x)

( )

Comments and Suggestions for Authors

This study aimed to study the relationship of free fatty acids (FFAs) and eicosanoids levels with the severity of depressive symptoms in stroke. Generally, the study raised an interesting and important point in stroke area. But there are some points that the authors should clarify before further consideration.

  1. The criteria for the inclusion of stroke patients should be placed. The authors only describe “The inclusion criterion was the ischemic stroke diagnosed on the basis of clinical symptoms and additional tests results, with standard treatment” in 4.1 Subjects section. More details should be added to avoid over-interpretation.

The following excerpts have been added to 4.1 Subjects section.

“The inclusion criterion was the ischemic stroke diagnosed on the basis of clinical symptoms and additional tests results, with standard treatment and brain imaging (computed tomography or nuclear magnetic resonance) [39,40]. Patients with both atherothrombotic and embolic stroke were also enrolled in the study. Stroke was defined as a syndrome of rapidly developing symptoms of focal or global cerebral dysfunction lasting ≥24 hours or leading to death, with apparent vascular causa [Hatano S. Experience from a multicentre stroke register: a preliminary report. Bull World Health Organ 1976; 54: 541-53]. The stroke aetiology was classified according to the TOAST classification system [Amarenco P, Bogousslavsky J, Caplan LR, Donnan GA, Hennerici MG. Classification of stroke subtypes. Cerebrovasc Dis 2009; 27: 493-501]. The exclusion criteria included: intracranial haemorrhage visible in brain imaging, symptoms of active infection including body temperature of more than 37.4°C, clinical or biochemical symptoms of infection, active autoimmune disorder or malignancy as well as the speech or consciousness disturbances to enable reliable results of BDI-II testing.”

“Seven patients received intravenous thrombolytic infusion of alteplase, 11 patients were taking L-thyroxine, 61 patients were taking hipotensives, 17 were on hypoglicemics. All patients received acetylsalicylic acid and statins.”

“All the subjects were informed of the aim of the study and signed a written consent to participate in the project.”

  1. The depression symptom may be affected by stroke type. For example, Toast classification, stroke of atherosclerosis or cardioembolism? Would that influence the results? The authors should give subgroup analysis?

Following the Reviewer’s suggestion, the subgroup analysis has been added to the results section. The appropriate tables including the analysis of TOAST classification with FFA and eicosanoids were added to the section “Results” (Table 7 and 8). These results were discussed and the references were included in the Discussion section as the following content:

“We evaluated the association between the types of stroke according to the TOAST classification system, eicosanoids and FFAs. The level of maresin 1, which belongs to the SPM, was higher in embolic strokes compared to the large-artery subtype. Maresin 1 attenuates inflammation in endothelial cells and prevents progression of atherosclerosis, however, no relevant research is available to compare [84]. The proinflammatory lipid mediator leukotriene B4 reached a higher level in small-vessel type of stroke compared to a large-vessel one. The genetic variant of the leukotriene B4 receptor complex was shown to affect the risk of cardioembolic stroke [85].  The bioactive form of lipoxin A4 - 15-epi-lipoxin A4 (15-epi LXA4) that promotes resolution of inflammation by inhibiting superoxide generation and polymorphonuclear leukocyte transmigration was found to be higher in embolic when compared to undetermined subtype of stroke [86]. It was observed that higher levels of SFA and n6 PUFA were associated with lacunar and atherosclerotic stroke, while no relationships regarding the n3 DHA and the subtypes of stroke were observed [87,88]. Higher levels of C14:0 myristic acid were identified in small-vessel subtypes of stroke in comparison to large-vessel and embolic ones, but in other SFAs (C15:0 pentadecanoid acid) its level was lower in small-vessel compared to undetermined type. We observed consistent results in regard to n6 FFA – lower levels were detected in embolic strokes compared to large and small-vessel subtypes. These finding are all the more interesting that C20:3n6 eicosatrienoic acid, also known as dihomo-γ-linolenic acid (DGLA) is engaged in anti-inflammatory, anti-platelet, anti-thrombotic and anti-atherogenic activities [89,90]. There is no direct data regarding the subtypes of stroke and the topic of our study. Bearing in mind the immunomodulating effects of certain FFAs and eicosanoids, we cannot exclude that the results obtained in the TOAST analysis may be of importance in the theory of common inflammatory background and interactions between lipids, their metabolites, stroke and depression. Such aspects need further studies which should aim at assessing the role of FFAs and eicosanoids in the stroke pathomechanism.”

Table 7. The comparison of subgroups (n =74) in regard to TOAST classification and FFA.

FFA [%]

1 vs. 2

1 vs. 3

1 vs. 5

2 vs. 3

2 vs. 5

3 vs. 5

C13:0 tridecanoic acid

NS

NS

NS

NS

NS

NS

C14:0 myristic acid

NS

0.033

NS

0.041

NS

NS

C14:1 myristolenic acid

NS

NS

NS

NS

NS

NS

C15:0 pentadecanoid acid

NS

NS

NS

NS

NS

0.021

C15:1 cis-10-pentadecanoid acid

NS

NS

NS

NS

NS

NS

C16:0 palmitic acid

NS

NS

NS

NS

NS

NS

C16:1 palmitoleic acid

NS

NS

NS

NS

NS

NS

C17:0 heptadecanoic acid

NS

NS

NS

NS

NS

NS

C17:1 cis-10- heptadecanoid acid

NS

NS

NS

NS

NS

NS

C18:0 stearic acid

NS

NS

NS

NS

NS

NS

C18:1n9 ct oleic acid

NS

NS

NS

NS

NS

NS

C18:1 vaccinic acid

NS

NS

NS

NS

NS

NS

C18:2n6c linoleic acid

NS

NS

NS

NS

NS

NS

C18:2n6t linoleic acid

NS

NS

NS

NS

NS

NS

C18:3n6 gamma linoleic acid

0.017

NS

0.037

0.025

NS

NS

C18:3n3 linolenic acid

NS

NS

0.006

NS

NS

NS

C18:4 stearidonate

NS

NS

NS

NS

NS

NS

C20:0 arachidic acid

NS

NS

NS

NS

NS

0.036

C22:1/C20:1 Cis11- eicosanic acid

NS

NS

NS

NS

NS

NS

C20:2 cis-11-eicodienoic acid

NS

NS

NS

NS

NS

NS

C20:3n6 eicosatrienoic acid

<0.001

NS

NS

0.007

0.007

NS

C20:4n6 arachidonic acid

NS

NS

NS

NS

NS

NS

C20:3n3 Cis-11-eicosatrienoic acid

NS

NS

NS

NS

NS

NS

C20:5n3 eicosapentaenoic acid

NS

NS

NS

NS

NS

0.034

C22:0 behenic acid

NS

NS

NS

NS

NS

0.034

C22:1n9 13 erucic acid

NS

NS

NS

NS

NS

NS

C22:2 cis-docodienoic acid

NS

NS

NS

NS

NS

0.039

C23:0 tricosanoic acid

NS

NS

NS

NS

NS

NS

C22:4n6 docosatetraenoate

NS

NS

NS

NS

NS

NS

C22:5w3 ddocosapentaenate

NS

NS

NS

NS

NS

NS

C24:0 lignoceric acid

NS

NS

NS

NS

NS

0.034

C22:6n3 docosahexaenoic acid

NS

NS

NS

NS

NS

NS

C24:1 nervonic acid

NS

NS

NS

NS

NS

NS

NS statistically non-significant differences, P< 0.05 statistically significant differences, TOAST classification: 1 large-artery atherosclerosis, 2 cardioembolism, 3 small vessel occlusion (lacunar), 5 undetermined cause.

Table 8. The comparison of subgroups (n =74) in regard to TOAST classification and eicosanoids.

Eicosanoids [µg/mL]

1 vs. 2

1 vs. 3

1 vs. 5

2 vs. 3

2 vs. 5

3 vs. 5

Resolvin E1

NS

NS

NS

NS

NS

NS

Prostaglandin E2

NS

NS

NS

NS

NS

NS

Resolvin D1

NS

NS

NS

NS

NS

NS

Lipoxin A4 LxA4 5S, 6R

NS

NS

NS

NS

NS

NS

Lipoxin A4 15-epi-LxA4 A4 5S, 6R, 15R

NS

NS

NS

NS

0.028

0.011

Protectin D1

NS

NS

NS

NS

NS

NS

Maresin 1

0.019

NS

NS

NS

NS

NS

Leucotriene B4

NS

0.037

NS

NS

NS

NS

18RS HEPE

NS

NS

NS

NS

NS

NS

16RS HETE

NS

NS

NS

NS

NS

NS

13S HODE

NS

NS

NS

NS

NS

NS

9S HODE

NS

NS

NS

NS

NS

NS

15S HETE

0.011

NS

NS

NS

NS

NS

17RS HDHA

NS

NS

NS

NS

NS

NS

12S HETE

NS

NS

NS

NS

NS

NS

5-oxo-ETE

NS

NS

0.045

NS

NS

NS

5 HETE

NS

NS

NS

NS

NS

NS

NS statistically non-significant differences, P< 0.05 statistically significant differences, TOAST classification: 1 large-artery atherosclerosis, 2 cardioembolism, 3 small vessel occlusion (lacunar), 5 undetermined cause.

  1. NIHSS is a commonly used scale to assess stroke severity. Patients with severe symptom may be more likely to fall into a depressive state. And did they get any improvement during follow-up? The authors should clarify this.

We did not assess patients with the use of NIHSS scale in the follow-up period of 6 months. Thank you for this suggestion. It would have  been interesting to explore this aspect  to find out if there is an association between the change in severity of the depressive symptoms and neurological deficit after 6 months. This is a disadvantage of our study, which we have included as  6. Limitations section:

The limitation of our study includes the fact that we performed the follow-up examination of subjects after 6 months in regard to the BDI-II scale but we did not provide follow-up assessment with the NIHSS scale. The severity and risk of depression may be related to the degree of neurological deficit. The improvement in the functional outcome could affect the BDI-II score in the follow-up period. It would be interesting to assess such potential relationships.

  1. More “standard treatment” information should be provided.

The information concerning “standard treatment” has been provided in the methods section.

Reviewer 2 Report

The submission from Dariusz Kotlega et al. report the association of Free Fatty Acids and Eicosanoids with the Depressive Symptoms typically of the Stroke in clinical study. They suggest that diet-derived FFAs affected the inflammatory pathways in pathogenesis of depression in stroke and reduced DHA levels attenuating depressive symptoms in stroke patients. The manuscript is interesting but some corrections are needed:

  1. Authors should highlight the novelty of the study.

  1. The authors should better check the manuscript for any typographical errors.

Author Response

Review 2

We greatly appreciate your time and effort dedicated to providing feedback on our manuscript and we are grateful for the insightful comments on and valuable improvements to our paper. All the suggestions helped us to evaluate our outcomes even more precisely in order to deliver improved, high quality scientific manuscript which we hope will now meet the high standards of  International Journal of Molecular Sciences.

Open Review

English language and style

( ) Extensive editing of English language and style required 
( ) Moderate English changes required 
(x) English language and style are fine/minor spell check required 
( ) I don't feel qualified to judge about the English language and style

Yes

Can be improved

Must be improved

Not applicable

Does the introduction provide sufficient background and include all relevant references?

(x)

( )

( )

( )

Is the research design appropriate?

(x)

( )

( )

( )

Are the methods adequately described?

(x)

( )

( )

( )

Are the results clearly presented?

( )

(x)

( )

( )

Are the conclusions supported by the results?

(x)

( )

( )

( )

Comments and Suggestions for Authors

The submission from Dariusz Kotlega et al. report the association of Free Fatty Acids and Eicosanoids with the Depressive Symptoms typically of the Stroke in clinical study. They suggest that diet-derived FFAs affected the inflammatory pathways in pathogenesis of depression in stroke and reduced DHA levels attenuating depressive symptoms in stroke patients. The manuscript is interesting but some corrections are needed:

  1. Authors should highlight the novelty of the study.

As suggested by the Reviewer, 5. Conclusions section has been corrected and supplemented with the following content:

“While emphasising the importance of nutrition itself, we also contributed to the nutritional and subsequently inflammatory theory of depression development in general population and  in stroke patients in particular. For the first time we elucidated a potential common association between dietary and inflammatory factors as those that affect the risk of depression with regard to the inflammatory theory of stroke pathomechanism. The severity of depressive symptoms, measured by means of BDI-II scale, is clearly related to the arachidonic acid cascade directly after the stroke onset and 6 months later. Moreover, the low level of DHA, in particular PUFA n3, can be a potential factor affecting the risk of depression and severity of depressive symptoms in stroke patients.”

  1. The authors should better check the manuscript for any typographical errors.

While we appreciate the Reviewer’s feedback, we had to comply with the layout and editing requirements of the journal. However, the manuscript has been proofread for a consistent use of UK spelling, punctuation and indented lines. The sequence and consecutive numbers of all the sections have been double checked.

Round 2

Reviewer 1 Report

Most of my questions are answered. I think current version is ready for acceptance.